# STEERABLE CNNs

**Taco S. Cohen**
University of Amsterdam
`t.s.cohen@uva.nl`

**Max Welling**
University of Amsterdam
Canadian Institute for Advanced Research
`m.welling@uva.nl`

## ABSTRACT

It has long been recognized that the invariance and equivariance properties of a representation are critically important for success in many vision tasks. In this paper we present Steerable Convolutional Neural Networks, an efficient and flexible class of equivariant convolutional networks. We show that steerable CNNs achieve state of the art results on the CIFAR image classification benchmark. The mathematical theory of steerable representations reveals a *type system* in which any steerable representation is a composition of elementary *feature types*, each one associated with a particular kind of symmetry. We show how the parameter cost of a steerable filter bank depends on the types of the input and output features, and show how to use this knowledge to construct CNNs that utilize parameters effectively.

## 1 INTRODUCTION

Much of the recent progress in computer vision can be attributed to the availability of large labelled datasets and deep neural networks capable of absorbing large amounts of information. While many practical problems can now be solved, the requirement for big (labelled) data is a fundamentally unsatisfactory state of affairs. Human beings are able to learn new concepts with very few labels, and reproducing this ability is an important challenge for artificial intelligence research. From an applied perspective, improving the statistical efficiency of deep learning is vital because in many domains (e.g. medical image analysis), acquiring large amounts of labelled data is costly.

To improve the statistical efficiency of machine learning methods, many have sought to learn invariant representations. In deep learning, however, intermediate layers should not be invariant, because the relative pose of local features must be preserved for further layers (Cohen & Welling, 2016; Hinton et al., 2011). Thus, one is led to the idea of *equivariance*: a network is equivariant if the representations it produces transform in a predictable way under transformations of the input. In other words, equivariant networks produce representations that are *steerable*. Steerability makes it possible to apply filters not just in every *position* (as in a standard convolution layer), but in every *pose*, thus allowing for increased parameter sharing.

Previous work has shown that equivariant CNNs yield state of the art results on classification tasks (Cohen & Welling, 2016; Dieleman et al., 2016), even though they only enforce equivariance to small groups of transformations like rotations by multiples of 90 degrees. Learning representations that are equivariant to larger groups is likely to result in further gains, but the computational cost of current methods scales linearly with the size of the group, making this impractical. In this paper we present the general theory of steerable CNNs, which covers previous approaches but also shows how the computational cost can be decoupled from the size of the symmetry group, thus paving the way for future scaling.

To better understand the structure of steerable representations, we analyze them mathematically. We show that any steerable representation is a composition of low-dimensional elementary *feature types*. Each elementary feature can be steered independently of the others, and captures a distinct characteristic of the input that has an invariant or "objective" meaning. This doctrine of "observer-independent quantities" was put forward by (Weyl, 1939, ch. 1.4) and is used throughout physics. It has been applied to vision and representation learning by Kanatani (1990); Cohen (2013).

The mentioned type system puts constraints on the network weights and architecture. Specifically, since an equivariant filter bank is required to map given input feature types to given output feature types, the number of parameters required by such a filter bank is reduced. Furthermore, by the same logic that tells us not to add meters to seconds, steerability considerations prevent us from adding features of different types (e.g. for residual learning (He et al., 2016a)).

The rest of this paper is organized as follows. The theory of steerable CNNs is introduced in Section 2. Related work is discussed in Section 3, which is followed by classification experiments (Section 4) and a discussion and conclusion in Section 5.

## 2 STEERABLE CNNs

### 2.1 FEATURE MAPS AND FIBERS

Consider a 2D signal $f : \mathbb{Z}^2 \to \mathbb{R}^K$ with $K$ channels. The signal may be an input to the network or a feature representation computed by a CNN. Since signals can be added and multiplied by scalars, the set of signals of this signature forms a linear space $\mathcal{F}$. Each layer of the network has its own feature space $\mathcal{F}_l$, but we will often suppress the layer index to reduce clutter.

It is customary in deep learning to describe $f \in \mathcal{F}$ as a *stack* of feature maps $f_k$ (for $k = 1, \dots, K$). In this paper we also consider another decomposition of $\mathcal{F}$ into *fibers*. The fiber $F_x$ at position $x$ in the "base space" $\mathbb{Z}^2$ is the $K$-dimensional vector space spanned by all channels at position $x$. Thus, $f \in \mathcal{F}$ is comprised of *feature vectors* $f(x)$ that live in the fibers $F_x$ (see Figure 1(a)).

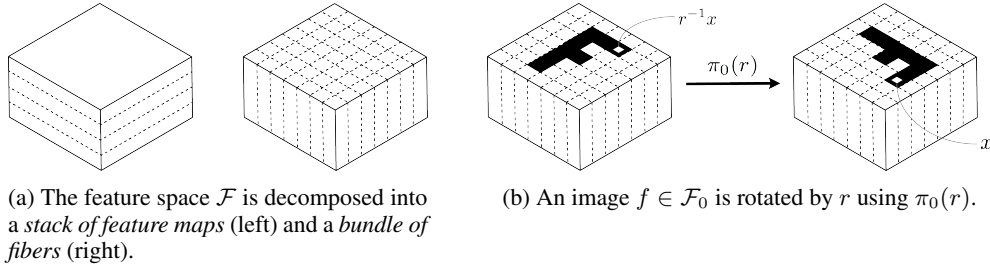

(a) The feature space $\mathcal{F}$ is decomposed into a *stack of feature maps* (left) and a *bundle of fibers* (right).

(b) An image $f \in \mathcal{F}_0$ is rotated by $r$ using $\pi_0(r)$.

Figure 1: Feature maps, fibers, and the transformation law $\pi_0$ of $\mathcal{F}_0$.

Given some group of transformations $G$ that acts on points in $\mathbb{Z}^2$, we can transform signals $f \in \mathcal{F}_0$:

$$[\pi_0(g)f](x) = f(g^{-1}x) \tag{1}$$

This says that the pixel at $g^{-1}x$ gets moved to $x$ by the transformation $g \in G$. We note that $\pi_0(g)$ is a linear operator.

An important property of $\pi_0$ is that $\pi_0(gh) = \pi_0(g)\pi_0(h)$. Here, $gh$ means composition of transformations in $G$, while $\pi_0(g)\pi_0(h)$ denotes matrix multiplication. A vector space such as $\mathcal{F}_0$ equipped with a set of linear operators $\pi_0$ satisfying this condition is known as a *group representation* (or just representation, for short). A lot is known about group representations (Serre, 1977), and we will make extensive use of the theory, explaining the relevant concepts as needed.

### 2.2 STEERABLE REPRESENTATIONS

Let $(\mathcal{F}, \pi)$ be a feature space with a group representation and $\Phi : \mathcal{F} \to \mathcal{F}'$ a convolutional network. The feature space $\mathcal{F}'$ is said to be (linearly) *steerable* with respect to $G$, if for all transformations $g \in G$, the features $\Phi f$ and $\Phi\pi(g)f$ are related by a linear transformation $\pi'(g)$ that does not depend on $f$. So $\pi'(g)$ allows us to "steer" the features in $\mathcal{F}'$ without referring to the input in $\mathcal{F}$ from which they were computed.

Combining the definition of steerability (i.e. $\Phi\pi(g) = \pi'(g)\Phi$) with the fact that $\pi$ is a group representation, we find that $\pi'$ must also be a group representation:

$$\pi'(gh)\Phi f = \Phi\pi(gh)f = \Phi\pi(g)\pi(h)f = \pi'(g)\Phi\pi(h)f = \pi'(g)\pi'(h)\Phi f \tag{2}$$

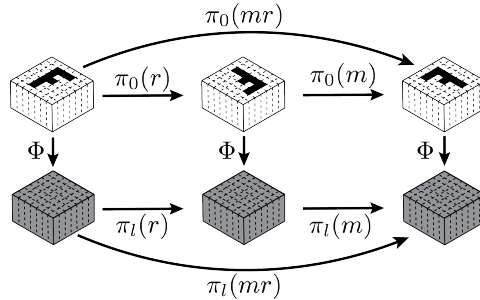

Figure 2: Diagram showing the structural consistency that follows from equivariance of the network $\Phi$ and the group representation structure of $\pi_0$. The result of following any path in this diagram depends only on the beginning and endpoint but is independent of the path itself, c.f. eq. 2

That is, $\pi'(gh) = \pi'(g)\pi'(h)$ (at least in the span of the image of $\Phi$). Figure 2 gives an illustration.

For simplicity, we will restrict our attention to discrete groups of transformations. The theory for continuous groups is almost completely analogous. Our running example will be the group $p4m$ which consists of translations, rotations by 90 degrees around any point, and reflections. We further restrict our attention to groups that are constructed[1] from the group of translations $\mathbb{Z}^2$ and a group $H$ of transformations that fixes the origin $\mathbf{0} \in \mathbb{Z}^2$. For $p4m$, we have $H = D4$, the 8-element group of reflections and rotations about the origin.

Using this division, we can first construct a filter bank that generates $H$-steerable fibers, and then show that convolution with such a filter bank produces a feature space that is steerable with respect to the whole group $G$.

## 2.3 EQUIVARIANT FILTER BANKS

A filter bank can be described as an array of dimension $(K', K, s, s)$, where $K, K'$ denote the number of input / output channels and $s$ is the kernel size. For our purposes it is useful to think of a filter bank as a linear map $\Psi : \mathcal{F} \to \mathbb{R}^{K'}$ that takes as input a signal $f \in \mathcal{F}$ and produces a $K'$-dimensional feature vector. The filter bank only looks at an $s \times s$ patch in $\mathcal{F}$, so the matrix representing $\Psi$ has shape $K' \times K \cdot s^2$. To correlate a signal $f$ using $\Psi$, one would simply apply $\Psi$ to translated copies of $f$, producing the output signal one fiber at a time.

We assume (by induction) that we have a representation $\pi$ that allows us to steer $\mathcal{F}$. In order to make the output of the convolution steerable, we need the filter bank $\Psi : \mathcal{F} \to \mathbb{R}^{K'}$ to be $H$-equivariant:

$$\rho(h)\,\Psi = \Psi\,\pi(h), \qquad \forall h \in H \tag{3}$$

for some representation $\rho$ of $H$ that acts on the output fibers (see Figure 3). Note that we only require equivariance with respect to $H$ (which excludes translations) and not $G$, because translations can move patterns into and out of the receptive field of a fiber, making full translation equivariance impossible.

The space of maps satisfying the equivariance constraint is denoted $\mathrm{Hom}_H(\pi, \rho)$, because an equivariant map $\Psi$ is a "homomorphism of group representations", meaning it respects the structure of the representations. Equivariant maps are also sometimes called *intertwiners* (Serre, 1977).

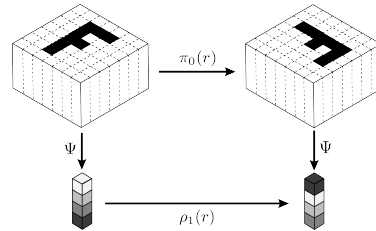

Figure 3: A filter bank $\Psi$ that is $H$-equivariant. In this example, $\rho_1$ represents the 90-degree rotation $r$ by a permutation matrix that cyclicly shifts the 4 channels.

Since the equivariance constraint (eq. 3) is linear in $\Psi$, the space $\mathrm{Hom}_H(\pi, \rho)$ of admissible filter banks is a vector space: any linear combination of maps $\Psi, \Psi' \in \mathrm{Hom}_H(\pi, \rho)$ is again an intertwiner. Hence, given $\pi$ and $\rho$, we can compute a basis for $\mathrm{Hom}_H(\pi, \rho)$ by solving a linear system.

---

[1] as a semi-direct product

Computation of the intertwiner basis is done offline, before training. Once we have such a basis $\psi_1, \ldots, \psi_n$ for $\mathrm{Hom}_H(\pi, \rho)$, we can express any equivariant filter bank $\Psi$ as a linear combination $\Psi = \sum_i \alpha_i \psi_i$ using parameters $\alpha_i$. As shown in Section 2.8, this can be done efficiently even in high dimensions.

## 2.4 INDUCTION

We have shown how to parameterize filter banks that intertwine $\pi$ and $\rho$, making the output fibers $H$-steerable by $\rho$ if the input space $\mathcal{F}$ is $H$-steerable by $\pi$. In this section we show how $H$-steerability of fibers $F'_x$ leads to $G$-steerability of the whole feature space $\mathcal{F}'$. This happens through a natural and important construction known as the *induced representation* (Mackey, 1952; 1953; 1968; Serre, 1977; Taylor, 1986; Folland, 1995; Kaniuth & Taylor, 2013).

As stated before, the correlation $\Psi \star f$ could be computed by translating $f$ before applying $\Psi$:

$$[\Psi \star f](x) = \Psi\left[\pi(x)^{-1}f\right]. \tag{4}$$

Where $x \in \mathbb{Z}^2$ is interpreted as a translation when given as input to $\pi$.

We can now calculate the transformation law of the output space. To do so, we apply a translation $t$ and transformation $r \in H$ to $f \in \mathcal{F}$, yielding $\pi(tr)f$, and then perform the correlation with $\Psi$. With a some algebra (Appendix A), we find:

$$[\Psi \star [\pi(tr)f]](x) = \rho(r)\left[[\Psi \star f]((tr)^{-1}x)\right] \tag{5}$$

Now if we define $\pi'$ as

$$[\pi'(tr)f](x) = \rho(r)\left[f((tr)^{-1}x)\right] \tag{6}$$

then $\Psi \star \pi(g)f = \pi'(g)\Psi \star f$ (see Fig. 4). This representation $\pi'$ is known as the representation of $G$ induced by the representation $\rho$ of $H$, and is denoted $\pi' = \mathrm{Ind}_H^G \rho$.

When parsing eq. 6, it is important to keep in mind that (as indicated by the square brackets) $\pi'$ acts on the whole feature space $\mathcal{F}'$ while $\rho$ acts on individual fibers.

If we compare the induced representation (eq. 6) to the representation $\pi_0$ defined in eq. 1, we see that the difference lies only in the presence of a factor $\rho(r)$ applied to the fibers. This factor describes how the feature channels are mixed by the transformation. The color channels in the input space do not get mixed by geometrical transformations, so we say that $\pi_0$ is induced from the trivial representation $\rho_0(h) = I$.

Now that we have a $G$-steerable feature space $\mathcal{F}'$, we can iterate the procedure by computing a basis for the space of intertwiners between $\pi'$ (restricted to $H$) and some $\rho'$ of our choosing.

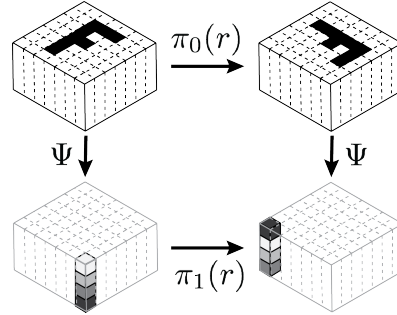

Figure 4: The representation $\pi_1$ induced from the permutation representation $\rho_1$ shown in fig. 3. A single fiber is highlighted. It is transported to a new location, and acted on by $\rho_1$.

## 2.5 FEATURE TYPES AND CHARACTER THEORY

By now, the reader may be wondering how to choose $\rho$, or indeed what the space of representations that we can choose from looks like in the first place. We will answer these questions in this section by showing that each representation has a *type* (encoded as a short list of integers) that corresponds to a certain symmetry or invariance of the feature. We further show how the number of parameters of an equivariant filter bank depends on the types of the representations $\pi$ and $\rho$ that it intertwines. Our discussion will make use of a number of important elementary results from group representation theory which are stated but not proved. The reader wishing to go deeper may consult chapters 1 and 2 of the excellent book by Serre (1977).

Recall that a group representation is a set of invertible linear maps $\rho(g) : \mathbb{R}^K \to \mathbb{R}^K$ satisfying $\rho(gh) = \rho(g)\rho(h)$ for all elements $g, h \in H$. It can be shown that any representation is a direct sum (i.e. `block_diag` plus change of basis) of a number of "elementary" representations associated with $G$. These building blocks are called irreducible representations (or irreps), because they can

| Irrep | Basis in $\mathcal{F}_0$ | | | $e$ | $r$ | $r^2$ | $r^3$ | $m$ | $mr$ | $mr^2$ | $mr^3$ |
|---|---|---|---|---|---|---|---|---|---|---|---|
| A1 |  |  |  | $[1]$ | $[1]$ | $[1]$ | $[1]$ | $[1]$ | $[1]$ | $[1]$ | $[1]$ |
| A2 | | | | $[1]$ | $[1]$ | $[1]$ | $[1]$ | $[-1]$ | $[-1]$ | $[-1]$ | $[-1]$ |
| B1 |  | | | $[1]$ | $[-1]$ | $[1]$ | $[-1]$ | $[1]$ | $[-1]$ | $[1]$ | $[-1]$ |
| B2 |  | | | $[1]$ | $[-1]$ | $[1]$ | $[-1]$ | $[-1]$ | $[1]$ | $[-1]$ | $[1]$ |
| E |  | | | $\begin{bmatrix}1&0\\0&1\end{bmatrix}$ | $\begin{bmatrix}0&-1\\1&0\end{bmatrix}$ | $\begin{bmatrix}-1&0\\0&-1\end{bmatrix}$ | $\begin{bmatrix}0&1\\-1&0\end{bmatrix}$ | $\begin{bmatrix}-1&0\\0&1\end{bmatrix}$ | $\begin{bmatrix}0&1\\1&0\end{bmatrix}$ | $\begin{bmatrix}1&0\\0&-1\end{bmatrix}$ | $\begin{bmatrix}0&-1\\-1&0\end{bmatrix}$ |

Table 1: The irreducible representations of the roto-reflection group D4. This group is generated by 90-degree rotations $r$ and mirror reflections $m$, and has 5 irreps labelled A1, A2, B1, B2, E. Left: decomposition of $\pi_0$ (eq. 1) in the space $\mathcal{F}_0$ of $3\times3$ filters with one channel. This representation turns out to have type $(3, 0, 1, 1, 2)$, meaning there are three copies of A1, one copy of B1, one copy of B2, and two copies of the 2D irrep E (A2 does not appear). Right: the representation matrices of each irrep, for each element of the group D4. The reader may verify that these are valid representations, and that the characters (traces) are orthogonal.

themselves not be block-diagonalized. In other words, if $\varphi_i$ are the irreducible representations of $H$, any representation $\rho$ of $H$ can be written in block-diagonal form:

$$\rho(g) = A \begin{bmatrix} \varphi_{i_1}(g) & & \\ & \ddots & \\ & & \varphi_{i_n} \end{bmatrix} A^{-1} \tag{7}$$

for some basis matrix $A$, and some $i_k$ that index the irreps (each irrep may occur 0 or more times).

Each irreducible representation corresponds to a type of symmetry, as shown in table 1. For example, as can be seen in this table, the representations $B1$ and $B2$ represent the 90-degree rotation $r$ as the matrix $[-1]$, so the basis filters for these representations change sign when rotated by $r$. It should be noted that in the higher layers $l > 0$, elementary basis filters can look different because they depend on the representation $\pi_l$ that is being decomposed.

The fact that all representations can be decomposed into a direct sum of irreducibles implies that each representation has a basis-independent *type*: which irreducible representations appear in it, and with what multiplicity. For example, the input representation $\pi_0$ (table 1) has type $(3, 0, 1, 1, 2)$. This means that, for instance, $\pi_0(r)$ is block-diagonalized as:

$$A^{-1}\pi_0(r)A = \texttt{block\_diag}([1], [1], [1], [-1], [-1], [0 \quad -1; 1 \quad 0], [0 \quad 1; -1 \quad 0]). \tag{8}$$

Where the block matrix contains $(3, 0, 1, 1, 2)$ copies of the irreps $(A1, A2, B1, B2, E)$, evaluated at $r$ (see column $r$ in table 1). The change of basis matrix $A$ is constructed from the basis filters shown in table 1 (and the same $A$ block-diagonalizes $\pi_0(g)$ for all $g$).

So the most general way in which we can choose a representation $\rho$ is to choose multiplicities $m_i \geq 0$ and a basis matrix $A$. In Section 2.7 we will find that there is an important restriction on this freedom, which alleviates the need to choose a basis. The choice of multiplicities is then the only hyperparameter, analogous to the choice of the number of channels in an ordinary CNN. Indeed, the multiplicities determine the number of channels: $K = \sum_i m_i \dim \varphi_i$.

## 2.6 DETERMINING THE TYPE OF THE INDUCED REPRESENTATION

By choosing the type of $\rho$, we also determine the type of $\pi = \text{Ind}_H^G \rho$ (restricted to $H$), but what is it? Explicit formulas exist (Reeder (2014); Serre (1977)) but are rather complicated, so we will present a simple computational procedure that can be used to determine the type of any representation. This procedure relies on the *character* $\chi_\rho(g) = \text{Tr}(\rho(g))$ of the representation to be decomposed. The most important fact about characters is that the characters of irreps $\varphi_i, \varphi_j$ are orthogonal:

$$\langle \chi_{\varphi_i}, \chi_{\varphi_j} \rangle \equiv \frac{1}{|H|} \sum_{h \in H} \chi_{\varphi_i}(h)\chi_{\varphi_j}(h) = \delta_{ij}. \tag{9}$$

Furthermore, since the trace of a direct sum equals the sum of the traces (i.e. $\chi_{\rho \oplus \rho'} = \chi_\rho + \chi_{\rho'}$), and every representation $\rho$ is a direct sum of irreps, it follows that we can obtain the multiplicity of irrep $\varphi_i$ in $\rho$ by computing the inner product with the $i$-th character:

$$\langle \chi_\rho, \chi_{\varphi_i} \rangle = \langle \chi_{\oplus_j m_j \varphi_j}, \chi_{\varphi_i} \rangle = \left\langle \sum_j m_j \chi_{\varphi_j}, \chi_{\varphi_i} \right\rangle = \sum_j m_j \langle \chi_{\varphi_j}, \chi_{\varphi_i} \rangle = m_i \qquad (10)$$

So a simple dot product of characters is all we need to determine the type of a representation. As we will see next, the type of the input and output representation of a layer determines the parameter cost of that layer.

### 2.6.1 THE PARAMETER COST OF EQUIVARIANT CONVOLUTION LAYERS

Steerable CNNs use parameters much more efficiently than ordinary CNNs. In this section we show how the number of parameters required by an equivariant layer is determined by the feature types of the input and output space, and how the efficiency of a choice of feature types may be evaluated.

In section 2.3, we found that a filter bank $\Psi$ is equivariant if and only if it lies in the vector space called $\mathrm{Hom}_H(\pi, \rho)$. It follows that the number of parameters for such a filter bank is equal to the dimensionality of this space, $n = \dim \mathrm{Hom}_H(\pi, \rho)$. This number is known as the *intertwining number* of $\pi$ and $\rho$ and plays an important role in the theory of group representations.

As with multiplicities, the intertwining number is easily computed using characters. It can be shown (Reeder, 2014) that the intertwining number equals:

$$\dim \mathrm{Hom}_H(\pi, \rho) = \langle \chi_\pi, \chi_\rho \rangle. \qquad (11)$$

By linearity and the orthogonality of characters, we find that $\dim \mathrm{Hom}_H(\pi, \rho) = \sum_i m_i m_i'$, for representations $\pi, \rho$ of type $(m_1, \ldots, m_J)$ and $(m_1', \ldots, m_J')$, respectively. Thus, as far as the number of parameters of a steerable convolution layer is concerned, the only choice we have to make for $\rho$ is its type – a short list of integers $m_i$.

The efficiency of a choice of type can be assessed using a quantity we call the *parameter utilization*:

$$\mu = \frac{\dim \pi \cdot \dim \rho}{\dim \mathrm{Hom}_H(\pi, \rho)}. \qquad (12)$$

The numerator equals $s^2 K \cdot K'$: the number of parameters for a non-equivariant filter bank. The denominator equals the parameter cost of an equivariant filter bank with the same filter size and number of input/output channels. Typical values of $\mu$ in effective architectures are around $|H|$, e.g. $\mu = 8$ for $H = D4$. Such a layer utilizes its parameters 8 times more intensively than an ordinary convolution layer.

### 2.7 EQUIVARIANT NONLINEARITIES & CAPSULES

In the previous section we showed that only the basis-independent types of $\pi$ and $\rho$ play a role in determining the parameter cost of an equivariant filter bank. An equivalent representation $\rho'(g) = A\rho(g)A^{-1}$ will have the same type, and hence the same parameter cost as $\rho$. However, when it comes to nonlinearities, different bases behave differently.

Just like a convolution layer (eq. 3), a layer of nonlinearities must commute with the group action. An elementwise nonlinearity $\nu : \mathbb{R} \to \mathbb{R}$ (or more generally, a fiber-wise nonlinearity $\nu : \mathbb{R}^K \to \mathbb{R}^{K'}$) is admissible for an input representation $\rho$ if there exists an output representation $\rho'$ such that $\nu$ applied after $\rho$ equals $\rho'$ applied after $\nu$.

Since commutation with nonlinearities depends on the basis, we need a more granular notion than the feature type. We define a $\rho$-capsule as a (typically low-dimensional) feature vector that transforms according to a representation $\rho$ (we may also refer to $\rho$ as the capsule). Thus, while a capsule has a type, not all representations of that type are equivalent as capsules. Given a catalogue of capsules $\rho^i$ (for $i = 1, \ldots, C$) with multiplicities $m_i$, we can construct a fiber as a *stack of capsules* that is steerable by a block-diagonal representation $\rho$ with $m_i$ copies of $\rho^i$ on the diagonal.

Like the capsules of Hinton et al. (2011), our capsules encode the pose of a pattern in the input, and consist of a number of units (dimensions) that do not get mixed with the units of other capsules by symmetries. In this sense, a stack of capsules is *disentangled* (Cohen & Welling, 2014).

We have found a few simple types of capsules and corresponding admissible nonlinearities. It is easy to see that any nonlinearity is admissible for $\rho$ when the latter is realized by permutation matrices: permuting a list of coordinates and then applying a nonlinearity is the same as applying the nonlinearity and then permuting. If $\rho$ is realized by a signed permutation matrix, then $\mathrm{CReLU}(\alpha) = (\mathrm{ReLU}(\alpha), \mathrm{ReLU}(-\alpha))$ introduced by Shang et al. (2016), or any concatenated nonlinearity $\nu'(\alpha) = (\nu(\alpha), \nu(-\alpha))$, will be admissible. Any scale-free concatenated nonlinearity such as CReLU is admissible for a representation realized by monomial matrices (having the same nonzero pattern as a permutation matrix). Finally, we can always make a representation of a finite group *orthogonal* by a suitable choice of basis, which means that we can use any nonlinearity that acts only on the length of the vector.

For many groups, the irreps can be realized using signed permutation matrices, so we can use irreducible $\varphi_i$-capsules with concatenated nonlinearities such as CReLU. Another class of capsules, which we call quotient capsules, are naturally realized by permutation matrices, and are thus compatible with any nonlinearity. These are described in Appendix C.

## 2.8 COMPUTATIONAL EFFICIENCY

Modern convolutional networks often use on the order of hundreds of channels $K$ per layer Zagoruyko & Komodakis (2016). When using $3 \times 3$ filters, a filter bank can have on the order of $9K^2 \approx 10^6$ dimensions. The number of parameters for an equivariant filter bank is about $\mu \approx 10$ times smaller, but a basis for the space of equivariant filter banks would still be about $10^6 \times 10^5$, which is too large to be practical.

Fortunately, the block-diagonal structure of $\pi$ and $\rho$ induces a block structure in $\Psi$. Suppose $\pi = \mathrm{block\_diag}(\pi^1, \ldots, \pi^P)$ and $\rho = \mathrm{block\_diag}(\rho^1, \ldots, \rho^Q)$. Then an intertwiner is a matrix of shape $K' \times Ks^2$, where $K' = \sum_i \dim \rho^i$ and $Ks^2 = \sum_i \dim \pi^i$. This matrix has the following block structure:

$$\Psi = \begin{bmatrix} h_{11} \in \mathrm{Hom}_H(\rho^1, \pi^1) & \cdots & h_{1P} \in \mathrm{Hom}_H(\rho^1, \pi^P) \\ \vdots & \ddots & \vdots \\ h_{R1} \in \mathrm{Hom}_H(\rho^R, \pi^1) & \cdots & h_{RP} \in \mathrm{Hom}_H(\rho^R, \pi^P) \end{bmatrix} \tag{13}$$

Each block $h_{ij}$ corresponds to an input-output pair of capsules, and can be parameterized by a linear combination of basis matrices $\psi_k^{ij} \in \mathrm{Hom}_H(\rho^i, \pi^j)$.

In practice, we typically use many copies of the same capsule (say $n_i$ copies of $\rho^i$ and $m_j$ copies of $\pi^j$). Therefore, many of the blocks $h_{ij}$ can be constructed using the same intertwiner basis. If we order equivalent capsules to be adjacent, the intertwiner consists of "blocks of blocks". Each superblock $H_{ij}$ has shape $n_i \dim \rho^i \times m_j \dim \pi^j$, and consists of subblocks of shape $\dim \rho^i \times \dim \pi^j$.

The computation graph for an equivariant convolution layer is constructed as follows. Given a catalogue of capsules $\rho^i$ and corresponding post-activation capsules $\mathrm{Act}_\nu \rho^i$, we compute the induced representations $\pi^i = \mathrm{Ind}_H^G \mathrm{Act}_\nu \rho^i$ and the bases for $\mathrm{Hom}_H(\rho^i, \pi^j)$ in an offline step. The bases are stored as matrices $\psi^{ij}$ of shape $\dim \rho^i \cdot \dim \pi^j \times \dim \mathrm{Hom}_H(\rho^i, \pi^j)$. Then, given a list of input / output multiplicities $n_i, m_j$ for the capsules, a parameter matrix $\Theta^{ij}$ of shape $\dim \mathrm{Hom}_H(\rho^i, \pi^j) \times n_i m_j$ is instantiated. The superblocks $H_{ij}$ are obtained by a matrix multiplication $\psi^{ij} \Theta^{ij}$ plus reshaping to shape $\dim \rho^i \cdot \dim \pi^j \times n_i m_j$. Once all superblocks are filled in, the matrix $\Psi$ is reshaped from $K' \times Ks^2$ to $K' \times K \times s \times s$ and convolved with the input.

## 2.9 USING STEERABLE CNNS IN PRACTICE

A full understanding of the theory of steerable CNNs requires some knowledge of group representation theory, but using steerable CNN technology is not much harder than using ordinary CNNs. Instead of choosing a number of channels for a given layer, one chooses a list of multiplicities $m_i$ for each capsule in a library of capsules provided by the developer. To preserve equivariance, the activation function applied to a capsule must be chosen from a list of admissible nonlinearities for that capsule (which sometimes includes all nonlinearities). Finally, one must respect the type system and only add identical capsules (e.g. in ResNets). These constraints can all be checked automatically.

## 3 RELATED WORK

Steerable filters were first studied for applications in signal processing and low-level vision (Freeman & Adelson, 1991; Greenspan et al., 1994; Simoncelli & Freeman, 1995). More or less explicit connections between steerability and group representation theory have been observed by Lenz (1989); Koenderink & Van Doorn (1990); Teo (1998); Krajsek & Mester (2007). As we have tried to demonstrate in this paper, representation theory is indeed the natural mathematical framework in which to study steerability.

In machine learning, equivariant kernels were studied by Reisert (2008); Skibbe (2013). In the context of neural networks, various authors have studied equivariant representations. Capsules were introduced in Hinton et al. (2011), and significantly improved by Tieleman (2014). A theoretical account of equivariant representation learning in the brain is given by Anselmi et al. (2014). Group equivariant scattering networks were defined and studied by Mallat (2012) for compact groups, and by Sifre & Mallat (2013); Oyallon & Mallat (2015) for the roto-translation group. Jacobsen et al. (2016) describe a network that uses a fixed set of (possibly steerable) basis filters with learned weights. Lenc & Vedaldi (2015) showed empirically that convolutional networks tend to learn equivariant representations, which suggests that equivariance could be a good inductive bias.

Invariant and equivariant CNNs have been studied by Gens & Domingos (2014); Kanazawa et al. (2014); Dieleman et al. (2015; 2016); Cohen & Welling (2016); Marcos et al. (2016). All of these models, as well as scattering networks, implicitly use the *regular representation*: feature maps are (often implicitly) conceived of as functions on $G$, and the action of $G$ on the space of functions on $G$ is known as the regular representation (Serre (1977), Appendix B). Our work is the first to consider other kinds of equivariance in the context of CNNs.

The idea of adding a type system to neural networks has been explored by Olah (2015); Balduzzi & Ghifary (2016). We have shown that a type system emerges naturally from the decomposition of a linear representation of a mathematical structure (a group, in our case) associated with the representation learned by a neural network.

## 4 EXPERIMENTS

We implemented steerable CNNs in Chainer (Tokui et al., 2015) and performed experiments on the CIFAR10 dataset (Krizhevsky, 2009) to determine if steerability is a useful inductive bias, and to determine the relative merits of the various types of capsules. In order to run experiments faster, and to see how steerable CNNs perform in the small-data regime, we used only 2000 training samples for our initial experiments.

As a baseline, we used the competitive wide residual networks (ResNets) architecture (He et al., 2016a;b; Zagoruyko & Komodakis, 2016). We tuned the capacity of this network for the reduced dataset size and settled on a 20 layer architecture (three residual blocks per stage, with two layers each, for three stages with feature maps of size $32 \times 32$, $16 \times 16$ and $8 \times 8$, various widths). We compared the baseline architecture to various kinds of steerable CNN, obtained by replacing the convolution layers by steerable convolution layers. To make sure that differences in performance were not simply due to underfitting or overfitting, we tuned the width (number of channels, $K$) using a validation set. The rest of the training procedure is identical to Cohen & Welling (2016), and is fixed for all of our experiments.

We first tested steerable CNNs that consist entirely of a single kind of capsule. We found that architectures with only one type do not perform very well (roughly 30-40% error, vs. 30% for plain ResNets trained on 2k samples from CIFAR10), except for those that use the regular representation capsule (Appendix C), which outperforms standard CNNs (26.75% error). This is not too surprising, because many capsules are quite restrictive in the spatial patterns they can express. The strong performance of regular capsules is consistent with the results of Cohen & Welling (2016), and can be explained by the fact that the regular representation contains all other (irreducible and quotient) representations as subrepresentations, and can therefore learn arbitrary spatial patterns.

We then created networks that use a mix of the more successful kinds of capsules. After a few preliminary experiments, we settled on a residual network that uses one mix of capsules for the input and output layer of a residual block, and another for the intermediate layer. The first representation

| Net | Depth | Width | #Params | #Labels | Dataset | Test error |
|---|---|---|---|---|---|---|
| Ladder | 10 | 96 | | 4k | C10ss | 20.4 |
| steer | 14 | (280, 112) | 4.4M | 4k | C10 | 23.66 |
| steer | 20 | (160, 64) | 2.2M | 4k | C10 | 24.56 |
| steer | 14 | (280, 112) | 4.4M | 4k | C10+ | 16.44 |
| steer | 20 | (160, 64) | 2.2M | 4k | C10+ | 16.42 |
| ResNet | 1001 | 16 | 10.2M | 50k | C10+ | 4.62 |
| Wide | 28 | 160 | 36.5M | 50k | C10+ | 4.17 |
| Dense | 100 | 2400 | 27.2M | 50k | C10+ | 3.74 |
| steer | 26 | (280, 112) | 9.1M | 50k | C10+ | 3.74 |
| steer | 20 | (440, 176) | 16.7M | 50k | C10+ | 3.95 |
| steer | 14 | (400, 160) | 9.1M | 50k | C10+ | **3.65** |
| ResNet | 1001 | 16 | 10.2M | 50k | C100+ | 22.71 |
| Wide | 28 | 160 | 36.5M | 50k | C100+ | 20.50 |
| Dense | 100 | 2400 | 27.2M | 50k | C100+ | 19.25 |
| steer | 20 | (280, 112) | 6.9M | 50k | C100+ | 19.84 |
| steer | 14 | (400, 160) | 9.1M | 50k | C100+ | **18.82** |

Table 2: Comparison of results of steerable CNNs vs. previous state of the art methods. A plus (+) indicates modest data augmentation (shifts and flips). Width for steerable CNNs is reported as a pair of numbers, one for the input / output layer of a ResNet block, and one for the intermediate layer.

consists of quotient capsules: regular, qm, qmr2, qmr3 (see Appendix C) followed by ReLUs. The second consists of irreducible capsules: A1, A2, B1, B2, E(2x) followed by CReLUs. On CIFAR10 with 2k labels, this architecture works better than standard ResNets and regular capsules at $24.48\%$ error.

When tested on CIFAR10 with 4k labels (table 2), the method comes close to the state of the art in *semi-supervised* methods, that use additional unlabelled data (Rasmus et al., 2015), and better than transfer learning approaches such as DCGAN which achieves $26.2\%$ error (Radford et al., 2015). When tested on the full CIFAR10 and CIFAR100 dataset, the steerable CNN substantially outperforms the ResNet (He et al., 2016b) baseline and achieves state of the art results (improving over wide and dense nets (Zagoruyko & Komodakis, 2016; Huang et al., 2016)).

# 5    CONCLUSION & FUTURE WORK

We have presented a theoretical framework for understanding steerable representations in convolutional networks, and have shown that steerability is a useful inductive bias that can improve model accuracy, particularly when little data is available. Our experiments show that a simple steerable architecture achieves state of the art results on CIFAR10 and CIFAR100, outperforming recent architectures such as wide and dense residual networks.

The mathematical connection between representation learning and representation theory that we have established improves our understanding of the inner workings of (equivariant) convolutional networks, revealing the humble CNN as an elegant geometrical computation engine. We expect that this new tool (representation theory), developed over more than a century by mathematicians and physicists, will greatly benefit future investigations in this area.

For concreteness, we have used the group of flips and rotations by multiples of 90 degrees as a running example throughout this paper. This group already has some nontrivial characteristics (such as non-commutativity), but it is still small and discrete. The theory of steerable CNNs, however, readily extends to the continuous setting. Evaluating steerable CNNs for large, continuous and high-dimensional groups is an important piece of future work.

Another direction for future work is *learning* the feature types, which may be easier in the continuous setting because (for non-compact groups) the irreps live in a continuous space where optimization may be possible. Beyond classification, steerable CNNs are likely to be useful in geometrical tasks such as action recognition, pose and motion estimation, and continuous control tasks.

ACKNOWLEDGMENTS

We kindly thank Kenta Oono, Shuang Wu, Thomas Kipf and the anonymous reviewers for their feedback and suggestions. This research was supported by Facebook, Google and NWO (grant number NAI.14.108).

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

## APPENDIX A: INDUCTION

In this section we will show that a stack of feature maps produced by convolution with an $H$-equivariant filter bank transforms according to the induced representation. That is, we will derive eq. 5, repeated here for convenience:

$$[\Psi \star [\pi_l(tr)f]](x) = \rho_{l+1}(r)\left[[\Psi \star f]((tr)^{-1}x)\right] \tag{14}$$

In the main text, we mentioned that $x \in \mathbb{Z}^2$ can be interpreted as a point or as a translation. Here we make this difference explicit, by writing $x \in \mathbb{Z}^2$ for a point and $\bar{x} \in G$ for a translation. (The operation $\bar{\cdot}$ defines a *section* of the projection map $G \to \mathbb{Z}^2$ that forgets the non-translational part of the transformation (Kaniuth & Taylor, 2013)).

With this notation, the convolution is defined as:

$$[\Psi \star f](x) = \Psi\pi(\bar{x}^{-1})f \tag{15}$$

Although the induced representation can be described in a more general setting, we will use an explicit matrix representation of $G$ to make it easier to check our computations. A general element of $G$ is written as:

$$g = tr = \begin{bmatrix} I & T \\ 0 & 1 \end{bmatrix} \begin{bmatrix} R & 0 \\ 0 & 1 \end{bmatrix} = \begin{bmatrix} R & T \\ 0 & 1 \end{bmatrix} \tag{16}$$

Where $R$ is the matrix representation of $r$ (e.g. a $2 \times 2$ rotation / reflection matrix), and $T$ is a translation vector. The section we use is:

$$\bar{x} = \begin{bmatrix} I & x \\ 0 & 1 \end{bmatrix} \tag{17}$$

Finally, we will distinguish the action of $G$ on itself, written $gh$ for $g, h \in G$ (implemented as matrix-matrix multiplication) and its action on $\mathbb{Z}^2$, written $g \cdot x$ for $g \in G$ and $x \in \mathbb{Z}^2$ (implemented as matrix-vector multiplication by adding a homogeneous coordinate to $x$).

To keep notation uncluttered, we will write $\pi = \pi_l$ and $\rho = \rho_{l+1}$. In full detail, the derivation of the transformation law for the feature space induced by $\rho$ proceeds as follows:

$$\begin{aligned}
[\Psi \star [\pi(tr)f]](x) &= \Psi\pi(\bar{x}^{-1})\pi(tr)f \\
&= \Psi\pi(\bar{x}^{-1}tr)f \\
&= \Psi\pi(rr^{-1}\bar{x}^{-1}tr)f \\
&= \Psi\pi(r)\pi(r^{-1}\bar{x}^{-1}tr)f \\
&= \rho(r)\Psi\pi(r^{-1}\bar{x}^{-1}tr)f \\
&= \rho(r)\Psi\pi((r^{-1}t^{-1}\bar{x}r)^{-1})f \\
&= \rho(r)\Psi\pi\left(\overline{(tr)^{-1} \cdot x}^{-1}\right)f \\
&= \rho(r)[\Psi \star f]((tr)^{-1} \cdot x)
\end{aligned} \tag{18}$$

The last line is the result shown in the paper. The justification of each step is:

1. Definition of $\star$
2. $\pi$ is a homomorphism / group representation

3. $rr^{-1}$ is the identity, so can always multiply by it

4. $\pi$ is a homomorphism / group representation

5. $\Psi \in \text{Hom}_H(\pi, \rho)$ is equivariant to $r \in H$.

6. Invert twice.

7. $\overline{(tr)^{-1} \cdot x} = r^{-1}t^{-1}\overline{x}r$ can be checked by multiplying the matrices / vectors.

8. Definition of $\star$

The derivation above is somewhat involved and messy, so the reader may prefer to think geometrically (using the figures in the paper) instead of algebraically. This complexity is an artifact of the lack of abstraction in our presentation. The induced representation is really a very natural object to consider (abstractly, it is the "adjoint functor" to the restriction functor. A more abstract treatment of the induced representation can be found in Serre (1977); Mackey (1952); Reeder (2014). A treatment that is close to our own, but more general is the "alternate description" found on page 49 of Kaniuth & Taylor (2013).

## Appendix B: Relation to Group Equivariant CNNs

In this section we show that the recently introduced Group Equivariant Convolutional Networks (G-CNNs, Cohen & Welling (2016)) are a special kind of steerable CNN. Specifically, a G-CNN is a steerable CNN with *regular* capsules.

In a G-CNN, the feature maps (except those of the input) are thought of as functions $f : G \to \mathbb{R}^K$ instead of functions on the plane $f : \mathbb{Z}^2 \to \mathbb{R}^K$, as we do here. It is shown that the feature maps transform according to

$$\pi(g)f(h) = f(g^{-1}h). \tag{19}$$

This defines a linear representation of $G$ known as the *regular representation*. It is easy to see that the regular representation is naturally realized by permutation matrices. Furthermore, it is known that the regular representation of $G$ is induced by the regular representation of $H$. The latter is defined in Appendix C, and is what we refer to as "regular capsules" in the paper.

## Appendix C: Regular and Quotient Features

Let $H$ be a finite group. A subgroup of $H$ is a subset that is also itself a group (i.e. closed under composition and inverses). The (left) coset of a subgroup $K$ in $H$ are the sets $hK = \{hk|k \in K\}$. The cosets are disjoint and jointly cover the whole group $H$ (i.e. they partition $H$). The set of all cosets of $K$ in $H$ is denoted $H/K$, and is also called the quotient of $H$ by $K$.

The coset space caries a natural left action by $H$. Let $a, b \in H$, then $a \cdot bK = (ab)K$.

This action translates into an action on the space of functions on $H/K$. Let $\mathcal{Q}$ denote the space of functions $f : H/K \to \mathbb{R}$. Then we have the following representation of $H$:

$$\rho(a)f(bK) = f(a^{-1} \cdot bK). \tag{20}$$

The function $f$ attaches a value to every coset. The $H$-action permutes these values, because it permutes the cosets. Hence, $\rho$ can be realized by permutation matrices. For small groups the explicit computations can easily be done by hand, while for large groups this task can be automated.

In this way, we get one permutation representation for each subgroup $K$ of $H$. In particular, for the subgroup $K = \{e\}$ (the trivial subgroup containing only the identity $e$), we have $H/K \cong H$. The representation in the space of functions on $H$ is known as the "regular representation". Using such regular representations in a steerable CNN is equivalent to using the group convolutions introduced in Cohen & Welling (2016), so steerable CNNs are a strict generalization of G-CNNs. At the other extreme, we take $K = H$, which gives the quotient $H/K \cong \{e\}$, the trivial group, which gives the trivial representation $A1$.

For the roto-reflection group $H = D4$, we have the following subgroups and associated quotient features

| Subgroup $K$ | quotient feature name | dimensionality |
|:---:|:---:|:---:|
| $\{e\}$ | regular | 8 |
| $\{e, m\}$ | qm | 4 |
| $\{e, mr\}$ | qmr | 4 |
| $\{e, mr^2\}$ | qmr2 | 4 |
| $\{e, mr^3\}$ | qmr3 | 4 |
| $\{e, r^2\}$ | r2 | 4 |
| $\{e, r, r^2, r^3\}$ | r | 2 |
| $e, r^2, m, mr^2$ | r2m | 2 |
| $e, r^2, mr, mr^3$ | r2mr | 2 |
| $H$ | A1 | 1 |

