# Peer review of "Steerable CNNs"

_ICLR 2017 — accepted_

[Public Comment · Taco Cohen · 20 Nov 2016]
**Revision uploaded**

In response to the following comment from Kenta Oono, we have uploaded a new version of the paper.

"I saw your paper submitted to ICLR 2017, Steerable CNNs.

It is wonderful as it is theoretically well founded with the representation theory and also takes engineering easiness into consideration. Further, it achieves state of the art result in CIFAR datasets. So I would like to understand the paper in detail.

As I read the paper, I encountered the problem that I could not derive some of equations (specifically, (5) in the paper). [...]"

The problem he identified was due to a small mistake in eq. 4 (a missing inverse). In the new version we have fixed this and added an appendix where eq. 5 (the induced representation) is derived.

[Public Comment · Shuang Wu · 09 Dec 2016]
**a few minor notation corrections**

- In the sentence right about equation (1), isn't it better to write "... a linear operator \pi_0(g): F_0 \rightarrow F_0.. " since \pi_0 is clearly defined for the case l = 0 as indicated by its subscript.
- right before equation (3), isn't it better to write " ... we need the filter bank \Psi: F_l \rightarrow R^{K_{l+1}} ..." since F here is for layer l specifically.

I am not sure I am reading the most recent revision, so forgive me if these things have already been fixed.

Great paper, by the way.

Shuang

[Public Comment · Shuang Wu · 11 Dec 2016]
**a few questions**

Hi, Taco,
  Great paper, I have a few comments/questions:
- is there code available for this work?
- how difficult (time-consuming) to train steerable CNNs on dataset such as ImageNet?
- How does steerable CNN compare against common CNNs such as ResNet or Inception in well-benchmarked tasks such as image classification? I feel the experiment section can be strengthened by add such results.
- Is there any reason besides mathematical simplicity to choose p4m? what about other group in the wallpaper group set?
- What are possible huddles to extend it to continuous group? What if we want to just consider finite group such as D_n instead of continuous group such as SO(2)?

Thanks.

PS: In the sentence right after equation (6) there is an equation for \pi_{l+1}, there is g on the LHS, I assume g should also appear on the RHS?

[Official Review · AnonReviewer3 · rating 8 · confidence 3 · 16 Dec 2016]
**A good step towards understanding good inductive bias in neural networks**
originality 2 · clarity 3 · impact 5 · substance 4

This paper essentially presents a new inductive bias in the architecture of (convolutional) neural networks (CNN). The mathematical motivations/derivations of the proposed architecture are detailed and rigorous. The proposed architecture promises to produce equivariant representations with steerable features using fewer parameters than traditional CNNs, which is particularly useful in small data regimes. Interesting and novel connections are presented between steerable filters and so called “steerable fibers”. The architecture is strongly inspired by the author’s previous work, as well as that of “capsules” (Hinton, 2011). The proposed architecture is compared on CIFAR10 against state-of-the-art inspired architectures (ResNets), and is shown to be superior particularly in the small data regime. The lack of empirical comparison on large scale dataset, such as ImageNet or COCO makes this largely a theoretical contribution. I would have also liked to see more empirical evaluation of the equivariance properties. It is not intuitively clear exactly why this architecture performs better on CIFAR10 as it is not clear that capturing equivariances helps to classify different instances of object categories. Wouldn’t action-recognition in videos, for example, not be a better illustrative dataset?

[Official Review · AnonReviewer1 · rating 7 · confidence 4 · 17 Dec 2016]
**No Title**
clarity 3 · meaningful comparison 5

The authors propose a parameterization of CNNs that guarantees equivariance wrt a large family of geometric transformations.

The mathematical analysis is rigorous and the material is very interesting and novel. The paper overall reads well; there is a real effort to explain the math accessibly, though some small improvements could be made.

The theory is general enough to include continuous transformations, although the experiments are restricted to discrete ones. While this could be seen as a negative point, it is justified by the experiments, which show that this set of transformations is powerful enough to yield very good results on CIFAR.

Another form of intertwiner has been studied recently by Lenc & Vedaldi [1]; they have studied equivariance empirically in CNNs, which offers an orthogonal view.

In addition to the recent references on scale/rotation deep networks suggested below, geometric equivariance has been studied extensively in the 2000's; mentioning at least one work would be appropriate. The one that probably comes closest to the proposed method is the work by Reisert [2], who studied steerable filters for invariance and equivariance, using Lie group theory. The difference, of course, is that the focus at the time was on kernel machines rather than CNNs, but many of the tools and theorems are relatable.


Some of the notation could be simplified, to make the formulas easier to grasp on a first read:

Working over a lattice Z^d is unnecessarily abstract -- since the inputs are always images, Z^2 would make much of the later math easier to parse. Generalization is straightforward, so I don't think the results lose anything by it; and the authors go back to 2D latices later anyway.

It could be more natural to do away with the layer index l which appears throughout the paper, and have notation for current/next layer instead (e.g. pi and pi'; K and D instead of K_{l+1} and K_l).

In any case I leave it up to the authors to decide whether to include these suggestions on notation, but I urge them to consider them (or other ways to unburden notation).


A few minor issues: Some statements would be better supported with an accompanying reference (e.g. "Explicit formulas exist" on page 5, the introduction of intertwiners on page 3). Finally, there is a tiny mistake in the Balduzzi & Ghifary reference (some extra information was included as an author name).

[1] Lenc & Vedaldi, "Understanding image representations by measuring their equivariance and equivalence", 2015
[2] Reisert, "Group integration techniques in pattern analysis: a kernel view", 2008

[Official Review · AnonReviewer4 · rating 6 · confidence 3 · 18 Dec 2016]
**Review of "steerable cnns"**

This paper presents a theoretical treatment of transformation groups applied to convnets, and presents some empirical results showing more efficient usage of network parameters.

The basic idea of steerability makes huge sense and seems like a very important idea to develop.  It is also a very old idea in image processing and goes back to Simoncelli, Freeman, Adelson, as well as Perona/Greenspan and others in the early 1990s.  This paper approaches it through a formal treatment of group theory.  But at the end of the day the idea seems pretty simple - the feature representation of a transformed image should be equivalent to a transformed feature representation of the original image.  Given that the authors are limiting their analysis to discrete groups - for example rotations of 0, 90, 180, and 270 deg. - the formalities brought in from the group theoretic analysis seem a bit overkill.  I'm not sure what this buys us in the end.  it seems the real challenge lies in implementing continuous transformations, so if the theory could guide us in that direction it would be immensely helpful.

Also the description of the experiments is fairly opaque.  I would have a hard time replicating what exactly the authors did here in terms of implementing capsules or transformation groups.

[Final Decision · Program Chairs · 06 Feb 2017]
**ICLR committee final decision**

The AC fully agrees with reviewer #4 that the paper contains a bit of an overkill in formalism: A lot of maths whose justification is not, in the end, very clear. The paper probably has an important contribution, but the AC would suggest reorganizing and restructuring, lessening the excess in formalism. 

As for the PCs, while we believe extending the experiments would further support the claims made in the paper, overall we still believe this paper deserves to appear at the conference as a poster.